

# Assessment of global total column water vapor sounding using a spaceborne differential absorption radar

Luis Millán[1], Richard Roy[1], and Matthew Lebsock[1]

[1]Jet Propulsion Laboratory, California Institute of Technology, Pasadena, California, USA

**Correspondence:** L. Millán (lmillan@jpl.nasa.gov)

**Abstract.**

The feasibility of using a differential absorption radar (DAR) to retrieve total column water vapor from space is investigated. DAR combines at least two radar tones near an absorption line, in this case a water vapor line, to measure humidity information from the differential absorption "on" and "off" the line. From a spaceborne platform, DAR can be used to retrieve total column water vapor by measuring the differential reflection from the Earth's Surface. We assess the expected precision, yield, and potential biases of retrieved total column water vapor values by applying an end-to-end radar instrument simulator to near-global weather analysis fields collocated with CloudSat measurements. The approach allows us to characterize the DAR performance across a globally representative dataset of atmospheric conditions including clouds and precipitation as well as different surface types.

We assume a hypothetical spaceborne G-band radar with pulse compression orbiting the earth at 405 km with a 1 m antenna, equivalent to a footprint diameter of 850 m, and 500 m horizontal integration. The simulations include the scattering effects of rain, snow, as well as liquid and ice clouds, spectroscopic uncertainties, and uncertainties due to the initial assumed water vapor profile. Results indicate that, using two radar tones at 167 and 174.8 GHz with a transmit power of 20 W ensures that both pulses will reach the surface at least 70% of the time in the tropics and more than 90% of the time outside the tropics, and that total column water vapor can be retrieved with a precision better than 1.3 mm.

## 1 Introduction

Water vapor is one of the most important gases in the Earth atmosphere. Its relevance has led to the development of several techniques for measuring its vertical distribution as well as the vertically integrated atmospheric water vapor content, which is often referred to as precipitable water, total water vapor, integrated water vapor, integrated precipitable water vapor or total column water vapor. The Global Climate Observing System has highlighted the utility of total column water vapor observations, declaring it as an essential climate variable (GCOS-ECV, 2020), which the World Meteorological Organization defines as a physical, chemical, or biological variable that critically contributes to the characterization of Earth's climate.

Passive satellite total column water vapor retrievals typically use instruments that measure in the visible (e.g., Lang et al., 2007; Wang et al., 2014), near-infrared (e.g., Lindstrot et al., 2012; Diedrich et al., 2015; Nelson et al., 2016), infrared (e.g., Pougatchev et al., 2009; Bedka et al., 2010), and microwave spectral regions (e.g., Schluessel and Emery, 1990; Wentz, 1997).



However, each spectral region has its limitations: Visible and near-infrared measurements are limited to cloud-free daytime regions and to land areas where the relatively bright surface reflection increases the signal compared to the dark ocean surfaces; infrared and microwave measurements can operate only over ice-free oceans, where the surface emissivity is well characterized. Additionally, infrared measurements are not possible in the presence of clouds or rain.

5    This study explores the feasibility of using a differential absorption radar (DAR) to remotely measure total column water vapor under all sky conditions, for all surfaces types and during day and night. The DAR technique is analogous to the differential absorption lidar (DIAL) technique (e.g., Schotland, 1966; Browell et al., 1979; Wulfmeyer and Walther, 2001), but operates in the microwave or millimeter-wave regime. In essence, the difference between the radar reflectivity at two nearby frequencies, "on" and "off" an absorption line, can be related to the amount of absorbing gas between the radar and the scattering target, which in this case is the Earth surface.

Prior studies have shown that the DAR technique can be used to derive water vapor profiles using three frequencies around 22 GHz (Meneghini et al., 2005), two frequencies at around 10 and 94 GHz (Tian et al., 2007) or at 2.8 and 35 GHz (Ellis and Vivekanandan, 2010). However, these studies require distinct radar transmitters for each frequency, which complicates the DAR measurement due to independent system calibration and beam overlap issues. Lebsock et al. (2015), Millán et al. (2016) and Battaglia and Kollias (2019) explored the feasibility of water vapor profiling with a narrow-band transmitter on the wings of the 183 GHz water vapor line, in which case the DAR measurement can be made with a single transceiver, greatly simplifying the measurement interpretation.

With respect to remotely measuring total column water vapor, Lebsock et al. (2015) and Millán et al. (2016) assessed the feasibility of using two frequencies near the 183 GHz water absorption line, using large eddy simulations and global cloud observations from CloudSat (Stephens et al., 2002), respectively. These studies concluded that the DAR technique could provide nearly spatially continuous observations of total column water vapor, even under rainy conditions. However, both studies assumed a *constant* instrument error model, with a 0.16 dBZ radar precision, and around a -30 dBZ minimum detectable signal. Here we use a more realistic uncertainty model (described in section 2) that includes speckle noise. That is, that depends on the magnitude of the return power which in turn depends on the water vapor burden. Furthermore, their simulations explored frequencies prohibited by the Federal Communications Commission for space-borne transmission.

Here, we assume a frequency-chirp, pulsed radar system, similar to the proof of concept instrument described by Cooper et al. (2018) and by Roy et al. (2018). Currently, a similar airborne radar operating near 170 GHz is being developed and has recently been validated during a ground-based deployment (Roy et al., 2019). The motivation of this study is to investigate the transmit power necessary to achieve close to global total column water vapor measurements from a space platform given realistic instrument and orbital parameters, and using frequencies that could be available for space transmission.

## 2 Differential absorption radar theory

It has been shown (Lebsock et al., 2015; Millán et al., 2016), that the ratio of two surface returns can be used to estimate the total column water vapor. Nevertheless, for completeness, a description of the radar theory is summarized here. Neglecting



multiple scattering, the surface return power measured by a monostatic radar which transmits a power $P_\mathrm{T}$ at a given frequency $\nu$ is given by,

$$P_\mathrm{R}(\nu) = \frac{P_\mathrm{T}(\nu)G(\nu)^2\lambda^2\Omega(\nu)}{(4\pi)^3 r^2}\Upsilon^2(\nu)\sigma_0(\nu) \tag{1}$$

where $G(\nu)$ is the antenna gain, $r$ is the distance to the surface, $\Omega(\nu)$ is the integral of the normalized two-way antenna pattern,

$\sigma_0(\nu)$ is the normalized surface cross section. $\Upsilon^2(\nu)$ is the two-way transmission given by,

$$\Upsilon^2(\nu) = \exp\left(-2\int_0^r [\sigma_\mathrm{gas}(\nu,r') + \sigma_\mathrm{Pext}(\nu,r')]\,\mathrm{d}r'\right) \tag{2}$$

where $\sigma_\mathrm{gas}(\nu,r)$ represents the gaseous absorption coefficient and $\sigma_\mathrm{Pext}(\nu,r)$ the particulate extinction (the sum of absorption and scattering) coefficient along the radar path.

If two radar tones are measured simultaneously, the ratio of two surface returns can be expressed as,

$$\frac{P_\mathrm{R}(\nu_2)}{P_\mathrm{R}(\nu_1)} = \frac{C(\nu_2)}{C(\nu_1)}\frac{\Upsilon^2(\nu_2)}{\Upsilon^2(\nu_1)}\frac{\sigma_0(\nu_2)}{\sigma_0(\nu_1)} \tag{3}$$

where $C(\nu)$ is the radar system parameter given by the first term of equation 1, that is, $\frac{P_\mathrm{T}(\nu)G(\nu)^2\lambda^2\Omega(\nu)}{(4\pi)^3 r^2}$.

Assuming that the frequency dependence of $\sigma_\mathrm{Pext}(\nu,r)$ and $\sigma_0(\nu)$ is small relative to that of $\sigma_\mathrm{gas}(\nu,r)$, this equation becomes

$$\frac{P_\mathrm{R}(\nu_2)}{P_\mathrm{R}(\nu_1)} = \frac{C(\nu_2)}{C(\nu_1)}\exp\left(-2\int_0^r [\sigma_\mathrm{gas}(\nu_2,r') - \sigma_\mathrm{gas}(\nu_1,r')]\,\mathrm{d}r'\right) \tag{4}$$

which can be rewritten as,

$$\frac{P_\mathrm{R}(\nu_2)}{P_\mathrm{R}(\nu_1)} = \frac{C(\nu_2)}{C(\nu_1)}\exp\left(-2\int_0^r \rho(r')\sum_i v_i(r')\left[\kappa_i(\nu_2,r') - \kappa_i(\nu_1,r')\right]\,\mathrm{d}r'\right) \tag{5}$$

where $\rho(r)$ is the air density and the sum is over all the absorbers with mass extinction cross section $\kappa_i(\nu,r)$ and volume mixing ratio $v_i(r)$. If the radar tones are close to a strong absorption line, the associated gas dominates the absorption. For example, absorption due to water vapor dominates the atmospheric attenuation near $183\,\mathrm{GHz}$. In that scenario, the only unknowns remaining are pressure, temperature and water vapor mixing ratio. It follows that assuming a temperature and pressure profile

(for example, from reanalysis fields or a climatology), and a water vapor profile shape, it should be possible to retrieve total column water vapor from the ratio of two surface returns. The uncertainties associated with these assumed profiles will be discussed in section 5.2.

To explore the capabilities of this technique, the radar reflectivity uncertainty needs to be properly simulated. Here we assume a moving satellite platform with velocity $V$ and antenna diameter $D$. Following Roy et al. (2018) and Roy et al. (2019),

the uncertainty in the received power (see equation 1), assuming decorrelated pulses, is given by

$$\Delta P_\mathrm{R}(\nu)^2 = \frac{P_\mathrm{R}(\nu)^2}{N_p} + \frac{2P_\mathrm{N}P_\mathrm{R}(\nu)}{N_p} + \frac{2P_\mathrm{N}^2}{N_p} \tag{6}$$


where $N_p$ is the number of pulses used in each measurement of $P_{\mathrm{R}}(\nu)$. The first term is due to speckle noise, the second term is known as Townes noise (i.e., Townes and Geschwind, 1948; Pearson et al., 2008), and the last term is due to the instrument thermal noise $P_{\mathrm{N}}$. Speckle noise can be understood as the variation in backscatter from randomly distributed scatterers causing interference effects in the coherent measurement of the total returned electric field. In simple terms, Townes noise is due to the

cross term of the sum of the signal and noise voltages. The instrument thermal noise is determined by

$$P_{\mathrm{N}} = \frac{k_b T_{\mathrm{sys}}}{\tau} \tag{7}$$

where $k_b$ is the Boltzmann constant, $T_{\mathrm{sys}}$ is the system noise temperature, and $\tau$ is the chirp time which we fix according to the relation given by (Walsh, 1982),

$$\tau = \frac{D}{2V} \tag{8}$$

which ensures that sequential pulses are decorrelated.

The number of pulses is then determined by

$$N_p = \varsigma \frac{T}{\tau} \tag{9}$$

where $\varsigma$ is the duty cycle, assumed to be 0.25 using a chirped pulse, and $T$ is the total integration time available for each radar tone, estimated using,

$$T = \frac{\Delta L}{V N_T} \tag{10}$$

where $\Delta L$ is the desired along-track horizontal resolution and $N_T$ is the number of radar tones (e.g., at least two, for the online and the offline tones).

In this study we assume a 405 km orbit, a 1 m antenna diameter, radar tones at 167 and 174.8 GHz, a system temperature of 1800 K, a desired horizontal integration of 500 m, and transmit powers varying from 0.1 to 100 W. These assumptions result in

a footprint diameter of $\sim$850 m, a chirp time of 66$\mu$s, a total incoherent integration time per tone of 33 ms and 125 number of pulses. These radar characteristics are listed in Table 1 (which also includes the symbols used throughout this study). Further, we define the minimum detectable signal to be $P_{\mathrm{R}}(\nu) = P_{\mathrm{N}}$ (i.e. where a single pulse signal-to-noise ratio (SNR) is equal to 1). This determines the sensitivity of the DAR measurement system. Despite the relatively long chirp time, we do not simulated any sidelobes because, as demonstrated by RainCube (a Ka-band 6U cubesat radar with a 166$\mu$s pulse length), through an

optimal selection of pulse shape and digital processing, sidelobes can be suppressed to accurately measure the most relevant precipitation processes near the surface (Peral et al., 2019).

The Earth's surface is a bright target, meaning that the SNR should be large. In this scenario, $P_{\mathrm{R}}(\nu)$ is generally much larger than $P_{\mathrm{N}}$ and the last 2 terms of equation 6 (the contributions from the Townes noise and the instrument noise) are small. Thus, in high-SNR regimes the fractional uncertainty in the received power simply becomes $1/\sqrt{N_p}$.





## 3 Radar Instrument Simulator

Radar returns are simulated using the radiometric model described in Millán et al. (2014) and Millán et al. (2016). In short, radar reflectivities are estimated using the time-dependent two-stream approximation (Hogan and Battaglia, 2008), gaseous absorption is evaluated using the clear sky forward model for the EOS Microwave Limb Sounder (Read et al., 2004), hydrom-

eteors scattering properties are evaluated using Mie scattering theory (assuming spherical hydrometeors), and the surface cross section is calculated using a quasi-specular scattering model (Li et al., 2005) for the ocean surface. More details can be found in Table 2.

The hydrometeor fields used in this study are supplied by cloud/rain profiles observed by CloudSat. CloudSat is a NASA satellite carrying a 94 GHz profiling radar sensitive to both cloud and precipitation particles (Stephens et al., 2002). CloudSat

retrievals provide the hydrometeor information while spatially and temporally interpolated weather analysis provides the meteorological conditions. In particular, rain and snow profiles are taken from the 2C-RAINPROFILE (Lebsock and L'Ecuyer, 2011) products and liquid water content (LWC) and ice water content (IWC) from the 2B-CWCRO R04 (Austin and Stephens, 2001; Austin et al., 2009). Temperature, pressure, and water vapor are taken from the European Centre for Medium-Range Weather Forecasts auxiliary (ECMWF-aux) products (Cronk and Partain, 2017). Note that, to decrease the number of cal-

culations we subsample these fields, we only used every 50 CloudSat measurement. It is noted that non-uniformities within the beam at scales smaller than the CloudSat horizontal resolution are not included in these simulations. They will be better addressed using either high resolution ground-based data or high resolution atmospheric models in subsequent studies.

Figure 1 shows an example simulation at 167 and 174.8 GHz, which are the two frequencies used throughout this study. These frequencies are the extreme frequency values achievable due to international transmission restrictions (NTIA, 2015).

The observed scene is heavily overcast with snow aloft and rain beneath the freezing level. The simulation is repeated for two distinct water vapor burdens; a wet atmosphere as estimated from the weather analysis and a hypothetical dry atmosphere (with 44 and 4 mm of total column water vapor burden and 298 and 257 K surface temperature, respectively). As expected, the dry atmosphere simulations show considerably less attenuation, but nevertheless, the impact of the water vapor burden is clearly visible in the extra attenuation experienced by the 174.8 GHz radar tone compared to 167 GHz. As examples of this

burden, Figure 2 shows the spectral variation of the surface return for different total column water vapor burdens under clear sky conditions. The spectral contrast between 174.8 and 167 GHz varies from 0.1dB for no water vapor to 31 dB for 60,mm of total column water vapor.

We use a quasi-specular surface backscatter model over the oceans (Li et al., 2005) assuming monthly climatological values derived from the ERA-Interim reanalysis for near surface wind speed and sea surface temperature (Dee et al., 2011). Over land

surfaces there are no empirical models for the radar cross section at these frequencies. Therefore we use the observed cross sections from CloudSat to scale the ocean frequency dependence of the Li et al. (2005) model as follows,

$$\sigma_0(\nu)' = \sigma_0^c(\nu_{94})\psi \tag{11}$$

where $\sigma_0(\nu)'$ is the modified surface cross section at a given frequency, $\sigma_0^c(\nu_{94})$ is the measured CloudSat cross section, and $\psi$ is the ratio between the cross section simulated by the ocean back scatter model at the desired frequency and the cross section





simulated by the ocean back scatter model at 94 GHz (CloudSat's frequency). Under most wind, temperature, and salinity conditions, $\psi$ is $\sim$0.78 at 167 GHz and $\sim$0.76 at 174.8 GHz. We emphasize that this method is ad-hoc and does not properly account for the differences in the spectral dependence of the reflectance properties of land surfaces versus water surfaces. Nonetheless it allows us to examine the feasibility of the remote sensing method. As an example, Figure 3 shows maps of

the measured CloudSat cross section as well as the modeled surface cross section at 167 GHz for the simulations used in this study. As explained by Haynes et al. (2009), $\sigma_0^c(\nu_{94})$ displays large spatial variability over land, where $\sigma_0(\nu)$ depends on vegetation, soil moisture, surface slope, snow cover, etc; as opposed to over the oceans where it mostly depends upon the wind speed through its effect on the surface slope distribution. Although equation 11 approximates the frequency dependence of land surface backscatter using an ocean model, the land-specific frequency dependence is minimal and should have only a minor

effects on results. The impact of uncertainties on $\sigma_0(\nu)$ will be discussed in section 5.2.

Figure 4 shows maps of the DAR simulations used in this study. The top panel is an 8-day average (January 1-8, 2007) of the CloudSat ECMWF-aux total column water vapor to show the context of the simulations. The next two panels show the average effective surface cross section, that is $\sigma_0(\nu)\Upsilon(\nu)^2$, at 167 and 174.8 GHz. Lastly, the bottom panel shows the difference between the 174.8 and 167 GHz simulations. Note that the simulations also include the hydrometeor burden, that

is, the IWC, LWC, rain and snow found on each Cloudsat profile, which in principle is frequency-dependent. In these maps there are around 8000 simulations (we only used every 50 CloudSat measurements). As shown, the impact of the water vapor burden can be seen at both radar tones, however, at 174.8 GHz the radar signal has been considerably more attenuated than at 167 GHz (which is further from the absorption line). Furthermore, the effective cross section difference (174.8 - 167 GHz), equivalent to the surface radar power ratios, is clearly strongly correlated with the total column water vapor field.

## 4    Retrieval Methodology

The aims of this study are to (1) quantify the uncertainties in DAR retrievals of total column water vapor and (2) to explore the trade-offs between radar transmit power and sampling of the Earth's real world meteorological variability. To accomplish this goal we performed end-to-end retrieval simulations. The retrieval algorithm used is,

$$w_{i+1} = w_i + \frac{\partial \hat{y}(w, \mathbf{b})}{\partial w} \left[ y - \hat{y}(w_i, \mathbf{b}) \right] \tag{12}$$

where the total column water vapor, $w_i$, is computed by suitably integrating the vertical water vapor profile $\mathbf{x}_i$, and $y$ is determined by the ratios between surface radar returns at different frequencies, that is to say

$$y = \frac{P_{\mathrm{R}}(\nu_2)}{P_{\mathrm{R}}(\nu_1)} \tag{13}$$

and the simulated measurements, $\hat{y}(w_i, \mathbf{b})$, are given by,

$$\hat{y}(w_i, \mathbf{b}) = \frac{F_{\nu_2}(w_i, \mathbf{b})}{F_{\nu_1}(w_i, \mathbf{b})} \tag{14}$$

where $F$ is the radar forward model described in section 3 and $\mathbf{b}$ is comprised of forward model parameters that influence the simulated radar observations but are not retrieved. For example,these include spectroscopic parameters, profiles of temperature,



pressure, ice water content, liquid water content, rain or snow. The assumptions made in $\mathbf{b}$ contribute to the systematic errors in the estimates of the total column water vapor. In each iteration, $\partial \hat{y}(w, \mathbf{b})/\partial w$ is evaluated by finite differences by perturbing the entire water vapor profile, $\mathbf{x}_i$, by 1%. Lastly, after each iteration $\mathbf{x}_{i+1}$ is computed following,

$$\mathbf{x}_{i+1} = \frac{w_{i+1}}{w_i}\mathbf{x}_i \tag{15}$$

That is, the shape of the assumed water vapor profile does not change during the retrieval, it is simply scaled according to the total column water vapor retrieved.

The estimated precision (the error due to random noise affecting the instrument) in the retrieved total column water vapor, $w$, is given by,

$$\sigma_w^2 = \left[\frac{\partial \hat{y}(w, \mathbf{b})}{\partial w}\sigma_y\right]^2 \tag{16}$$

where $\sigma_y$ is given by,

$$\sigma_y^2 = \left[\frac{P_{\mathrm{R}}(\nu_2)}{P_{\mathrm{R}}(\nu_1)}\sqrt{\left(\frac{\Delta P_{\mathrm{R}}(\nu_2)}{P_{\mathrm{R}}(\nu_2)}\right)^2 + \left(\frac{\Delta P_{\mathrm{R}}(\nu_1)}{P_{\mathrm{R}}(\nu_1)}\right)^2}\right]^2 \tag{17}$$

which is simply the propagation of combining the individual errors of $P_{\mathrm{R}}(\nu_2)$ and $P_{\mathrm{R}}(\nu_1)$.

To perform end-to-end retrievals, we first need to defined a set of conditions regarded as truth as well as the radar characteristics. As detailed in section 3, these conditions were taken from Cloudsat measurements (IWC, LWC, rain, snow, temperature

and pressure) while the radar characteristics are listed in Table 1. With these atmospheric conditions and radar parameters, we compute synthetic radar returns to be used as measurements, that is, the synthetic radar returns are given by

$$P_{\mathrm{R}} = F(w_T, \mathbf{b_T}) \tag{18}$$

where $w_T$ is the true water vapor state as provided by the CloudSat-ECMWF product, and where $\mathbf{b_T}$ represents the rest of the atmospheric state (temperature, pressure, and hydrometeor profiles) also provided by the CloudSat-ECMWF product.

These synthetic radar returns are then run through the retrieval algorithm. For the retrieved profile shape, $\mathbf{x_0}$, a water vapor profile taken from a ERA-Interim monthly climatology was used. The iterative procedure stops when $|1 - \frac{w_{i+1}}{w_i}|$ is lower than 0.05, which is normally achieved within 2 iterations.

During these retrievals we assume perfect knowledge of the forward model parameters, that is, the simulated radar returns used during the retrieval are given by,

$$\hat{P_{\mathrm{R}}} = F(w_i, \mathbf{b_T}) \tag{19}$$

or in other words, the only variable changing between $y$ and $\hat{y}$ is the water vapor burden.

Sensitivity to assumed parameters is estimated using a perturbed set of synthetic radar returns following,

$$P_{\mathrm{R}}' = F(w_T, \mathbf{b}') \tag{20}$$





where $\mathbf{b}'$ represents the perturbed forward model parameter. Only one of the parameters is perturbed at a time; for instance, when computing the systematic uncertainty related to temperature, only the temperature values are perturbed, while the rest (IWC, LWC, rain, snow, particle size distributions, etc) are left unperturbed. Then, the retrieved total column water vapor using the perturbed measurements are compared to the retrieved values from an unperturbed run, i.e. using the measurements given by equation 18. The difference between the two retrieved total column water vapor is used a measure of the impact of a given systematic error source. Instrument noise is not added to any of these simulations because its impacts are studied through equation 16.

## 5 Results

First we will explore the precision, that is, the expected random error associated with the radar uncertainty described in section 2. Then we will explore potential systematic errors, such as the impact of not knowing the hydrometeor burden by assuming clear sky conditions throughout the forward model simulations, the impact of not knowing precisely the temperature and pressure by using climatological values, the impact of changing the initial assumed water vapor profile, the impact of the spectroscopy errors, and surface roughness uncertainties by using a constant value.

### 5.1 Precision

Figure 5-left shows maps of the total column water vapor precision (random error) assuming different transmit powers. White areas denote regions with no CloudSat measurements to initialize the simulations (i.e. the poles) or regions where the pulses (the simulated $P_R$'s used in $\mathbf{y}$) are attenuated beneath the noise floor (i.e. the tropics when using 0.1 W of transmit power). As shown even with just a transmit power of 0.1 W, errors are better than 1.2 mm throughout the globe except at the tropics where the radar tones are completely attenuated. With a transmit power greater than 20 W errors are mostly below 1.2 mm everywhere except in active deep convective regions such as the maritime continent. Further increasing the transmit power does little to improve the precision, because as we noted in section 2, in high SNR regimes, the fractional error in the measurement is largely determined by the number of uncorrelated pulses used. Figure 6 shows the cumulative SNR histogram for the 174.8 GHz surface returns. As can be seen, most of the time, the SNR is higher than 10 even when using just 0.1 W. The SNR for the 167 GHz radar tone is slightly better since it experiences less water vapor attenuation.

Instead, increasing the transmit power improves the yield, that is, the number of times both pulses reached the surface divided by the total number of simulations. Figure 5-right shows fractional yield maps for the same transmit powers as those shown in left panels. Overall, even with only 0.1 W transmit power, the yield is better than 0.7 throughout most of the globe except at the tropics where the yield sharply drops to zero. The yield improves drastically when using at least 10 W.

To complement these maps, Figure 7 shows the random errors and the fractional yield zonal averages. Outside the tropics, that is polewards of $30^o$S or $30^o$N, regardless of transmit power, random errors are generally below 1.2 mm with fractional yields better than 0.7. With a transmit power of at least 10 W the random errors are below 1 mm and the fractional yield improves to better than 0.9. In the tropics, using at least 20 W, the random errors are mostly below 1.3 mm with fractional





yields better than 0.7, improving to random errors mostly below 1.2 mm with fractional yields better than 0.8 when using at least 50 W. Under clear sky conditions, the random errors remain mostly the same. The yield, however, improves substantially. For example, in the tropics, for 20 W of transmit power, the yields becomes better than 0.85 (as opposed to better than 0.7) and for 50 W become better than 0.95 (as opposed to better than 0.8).

To date, passive microwave instruments have provided the benchmark for total column water vapor measurements. For example, the Advanced Microwave Scanning Radiometer (AMSR) instruments have an estimated error of ∼0.6 mm (Wentz and Meissner, 2000) for a native footprint of around 14 km by 8 km (Kawanishi et al., 2003). The precision of aggregated DAR total column water vapor measurements (the ones simulated here) matching such a footprint would be considerably better.

## 5.2 Systematic Uncertainties

Figure 8 shows zonal averages of eight potential systematic uncertainties sources. As explained in section 4, these systematic errors arise from the uncertainties in the ancillary knowledge used (including the spectroscopy uncertainties) throughout the retrievals. For example, as shown by equation 5 the uncertainties in the water vapor mass extinction cross section, $\kappa(\nu, r)$, will affect the estimated total column water vapor. Note that these systematic errors are independent of the transmit power as long as the surface return is not completely attenuated by the atmosphere. As such, these simulations were performed using 100 W

to evaluate them under most conditions, that is, we use 100 W because it has the better yield. The systematic error sources studied here are explained below:

- pT (pressure-temperature) Climatology: Errors associated with using climatological pressure and temperature conditions throughout the forward model end-to-end simulated retrievals as opposed to using the actual pressure and temperature conditions. The climatological values correspond to January 2007 ERA-Interim monthly mean values for the 12 UT

synoptic time. Simply, these errors evaluate the worst possible impact of not knowing precisely the temperature and pressure.

- Clear Sky: Errors associated with assuming clear sky conditions throughout the end-to-end retrieval simulations as opposed to using the actual hydrometeors conditions. In other words, having no hydrometeor information to constrain the retrievals. Note that these error estimates evaluate the additional frequency dependent attenuation imposed by the

hydrometeors. That is, it is assumed that the radar system is capable of identifying the surface return through coarse ranging capability.

- Assumed water vapor profile: Errors associated with using a different linearization water vapor profile in the end-to-end simulated retrievals. The assumed profiles are perturbed by up to 20% by layers. That is, we perturb the profile between 0-2, 2-4, 4-6, and 6-8 km individually and then aggregate the systematic uncertainty.

- Multiple scattering: Errors associated with simulating single scattering returns as opposed to multiple scattering ones.

- $H_2O$ 183 GHz line strength: Error associated with perturbing the 183 GHz water vapor line strength by 0.25% following the uncertainty described by Pickett et al. (1998).





- H$_2$O 183 GHz line width: Error associated with perturbing the 183 GHz water vapor line width by 4% following Bauer et al. (1989) and Goyette and Lucia (1990).

- H$_2$O continuum: Error associated with perturbing the H$_2$O continuum by 10% following Meshkov and De Lucia (2005).

- O$_2$ and N$_2$ continuum: Errors associated with perturbing the O$_2$ and N$_2$O continuum by 10% following Meshkov and De Lucia (2005).

- Surface roughness: Errors associated with using a constant 12 ms$^{-1}$ as opposed to a surface wind climatology.

As shown in Figure 8, all the potential systematic uncertainties are lower than 0.5 mm, except for the errors associated with H$_2$O 183 GHz line width which could be as big as 1.4 mm. As expected, this uncertainty is approximately 4% of the total column water vapor because a H$_2$O line width perturbation mostly equates to perturbing the measurement, that is the ratio of the surface radar returns at different frequencies, by the same amount. However, this type of bias should be easily corrected during a validation campaign since all retrievals will be off by the same *constant* amount. In all scenarios simulated here, the surface return dwarfed the multiple-scattered component of clouds and rain. That is, the systematic uncertainty induced by ignoring multiple scattering effects was negligible however this may be due to the "coarse" resolution of Cloudsat hydrometeors.

## 6 Summary

We have evaluated the precision, yield, and systematic uncertainties of a differential absorption radar to measure total column water vapor from space. This technique requires at least two radar tones near the 183 GHz water vapor absorption line ("on" and "off" the line) to infer the humidity burden between the radar and the surface. In this work, we used 167 and 174.8 GHz, the extremes of the frequency range of VIPR (Roy et al., 2019). Further, we assume an antenna diameter of 1 m, a horizontal integration of 500 m, an integration time of 33 ms, a system temperature of 1800 K, an orbit of 405 km, and transmit powers varying from 0.1 to 100 W.

We apply a radar instrument simulator to weather analysis fields colocated with CloudSat near-global measurements to simulate surface radar returns to be used as measurements in end-to-end retrievals. We use an iterative least-squares fit retrieval algorithm that allow us to quantify both the expected precision and the impact of potential systematic uncertainties upon the retrieved total column water vapor.

Systematic uncertainties related to the pressure and temperature, the hydrometeor burden, the initial guess, the water vapor line strength, the water vapor, O$_2$ and N$_2$ continuum, multiple scattering effects, and the magnitude of the surface winds could result in potential biases lower than 0.5 mm. Systematic uncertainties associated with the water vapor line width could be up to 1.4 mm. This approximately corresponds to 4% of the total column water vapor because a H$_2$O line width perturbation mostly equates to perturb the ratio of the surface radar returns by the same amount.

Precision and yield results can be summarized as follows:





- Outside the tropics, regardless of transmit power, random errors are generally below 1.2 mm with fractional yields better than 0.7. With a transmit power of at least 10 W the random errors are below 1 mm and the fractional yield improves to better than 0.9.

- In the tropics, using at least 20 W, the random errors are mostly below 1.3 mm with fractional yields better than 0.7, improving to mostly below 1.2 mm with fractional yields better than 0.8 when using at least 50 W.

These results suggest that at least 20 W of transmit power are needed to be able to measure total column water vapor globally with a reasonable yield. Output powers in the 10-100 W range would require additional research and development. DAR holds considerable potential as a technique to study the distribution of total column water vapor globally, that is, under most terrains and under most meteorological conditions, with considerably high horizontal resolution.

*Data availability.* The CloudSat dataset used in this manuscript can be found on the CloudSat data processing center website (http://www.cloudsat.cira.colostate.edu/order-data)

*Author contributions.* LFMV wrote the algorithm and carried out the analyses. RJR and MDL provided scientific expertise throughout all stages of the research.

*Competing interests.* There are no competing interests

*Acknowledgements.* The research was carried out at the Jet Propulsion Laboratory, California Institute of Technology, under a contract with the National Aeronautics and Space Administration. We thank Ken Cooper for his support throughout this research.



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





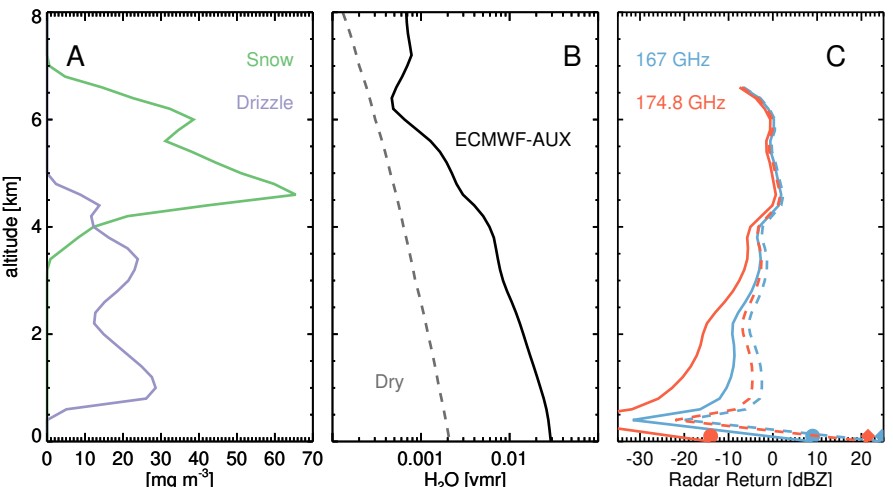

**Figure 1.** Example of CloudSat driven simulations. (A) Hydrometeor burden. (B) Water vapor burden for two scenarios: the ECMWF-AUX scenario (solid line) and dry case (dash line). (C) CloudSat driven simulations assuming the hydrometeor burden shown in panel A and the two water vapor burden scenarios (solid lines for the ECMWF-AUX scenario and dash lines for the dry case) shown in panel B for the two frequencies used in this study (half circles indicate the magnitude of the surface return in dBZ for the ECMWF-AUX scenario while triangles indicate it for the dry case).

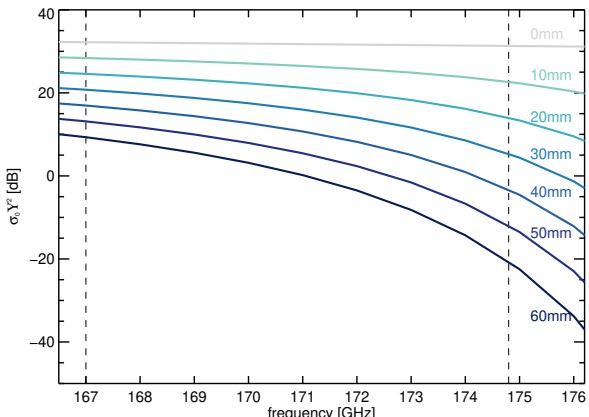

**Figure 2.** Examples of the spectral variation of the column surface return due to several total column water vapor burdens under clear sky conditions. Dashed vertical lines shows the two frequencies used in this study (167 and 174.8 GHz).



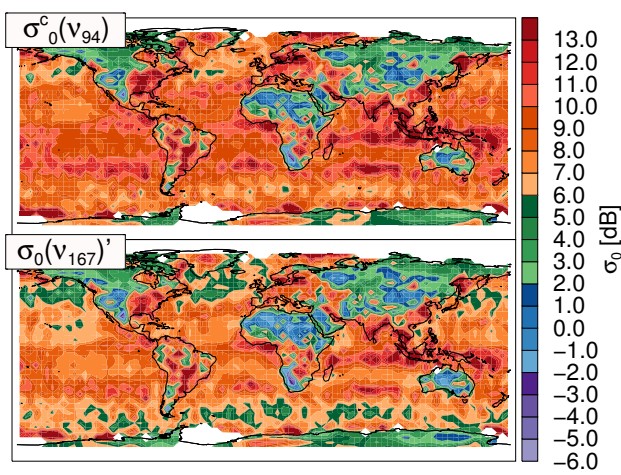

**Figure 3.** Mean CloudSat surface normalized backscattering cross section for January 1-8, 2007 ($\sigma_0^c(\nu_{94})$), and modified backscattering cross section at 167 GHz ($\sigma_0(\nu_{167})'$). Note that we are displaying the log of the average.

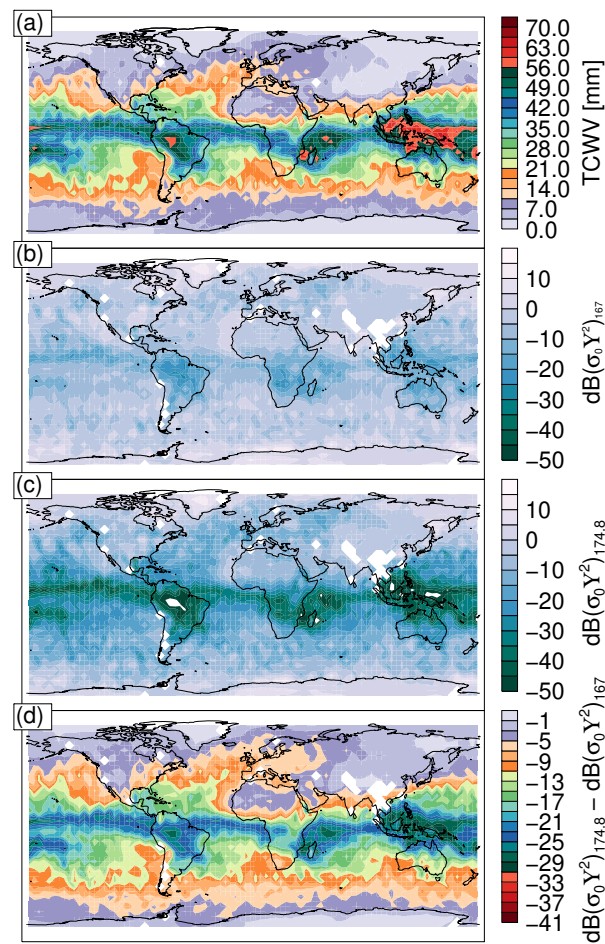

**Figure 4.** Maps exemplifying the CloudSat driven simulations burdens (January 1-8, 2007). (a) total column water vapor, (b) simulated CloudSat-driven effective surface cross-section at 167 GHz, (c) simulated CloudSat-driven effective surface cross-section at 174.8 GHz, and (d) effective surface cross section difference (174.8–167 GHz). Grid boxes are 4° longitude by 4° latitude.



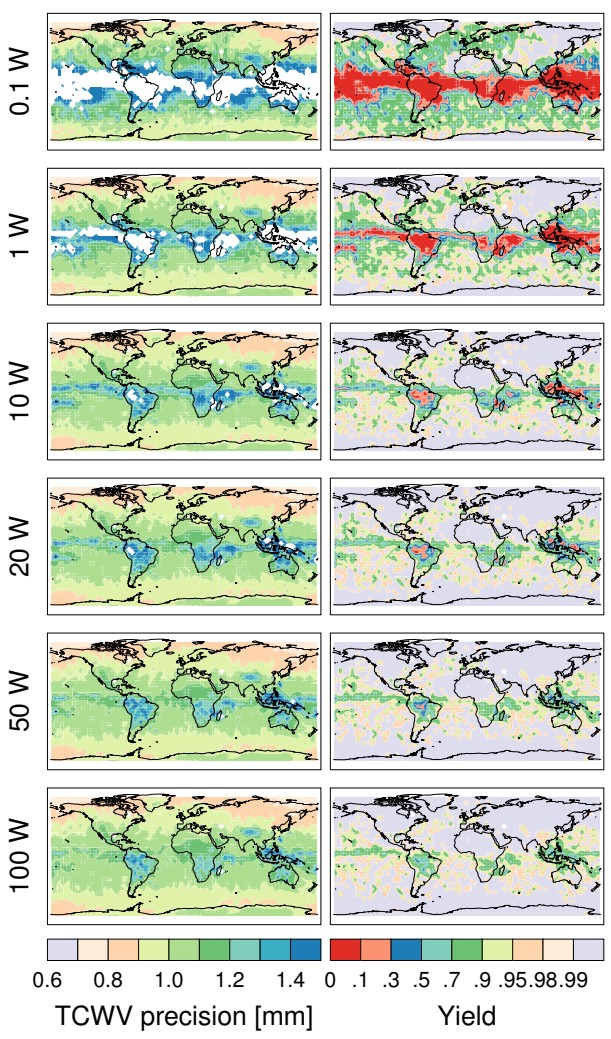

**Figure 5.** Total column water vapor (TCWV) precision maps as well as fractional yield maps for different transmit powers.

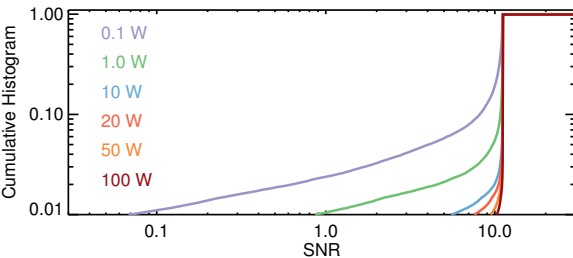

**Figure 6.** SNR cumulative histogram for the 174.8 GHz radar tone.





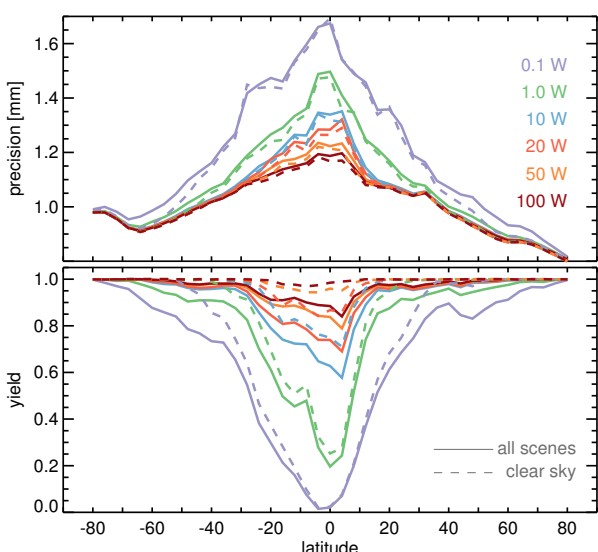

**Figure 7.** Total column water vapor precision (top) as well as fractional yield (bottom) for different transmit powers versus latitude.

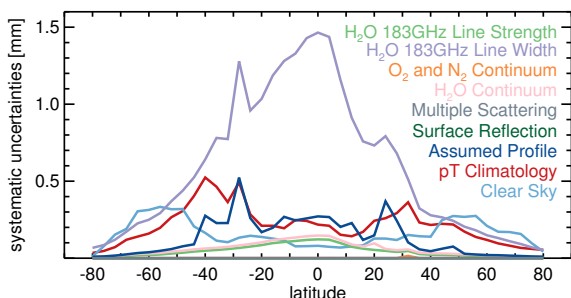

**Figure 8.** Systematic uncertainties versus latitude


**Table 1.** Satellite Radar Parameters

| Parameter | Symbol | Value | Units |
|---|---|---|---|
| Antenna Diameter | $D$ | 1 | m |
| Footprint Diameter | | $\sim$850 | m |
| Horizontal Resolution | $\Delta L$ | 500 | m |
| System Temperature | $T_{sys}$ | 1800 | K |
| Platform Altitude | $r$ | 405 | km |
| Platform Velocity | $V$ | 7669 | m/s |
| Duty cycle | $\varsigma$ | 0.25 | |
| Number of pulses | $N_P$ | 125 | |
| Chirp time | $\tau$ | 66 | $\mu$s |
| Integration time | T | 33 | ms |
| Transmit Powers | $P_T$ | 0.1, 1, 10, 20, 50, 100 | W |

**Table 2.** Radar Instrument Simulator Specifics

| Parameter | Detail |
|---|---|
| Water Dielectric Properties | Liebe et al. (1991) |
| Ice Dielectric Properties | Hufford (1991) |
| Ice Water Content (IWC) PSD[a] | McFarquhar and Heymsfield (1997) |
| Liquid Water Content (LWC) PSD | Using a log normal distribution with a 10 $\mu$m mean radius and a 1.3 spread. |
| Rain PSD | Abel and Boutle (2012) |
| Snow PSD | Sekhon and Sirvastava (1970) |
| Gas Absorption | Read et al. (2004) |
| Radiation Propagation | Hogan and Battaglia (2008); Hogan (2013) |
| Surface Reflection | Li et al. (2005) assuming climatological surface wind and skin temperature conditions, a Fresnel fraction of 1, and 35 ‰ parts per thousand salinity. |

[a] particle size distribution.