# Peer review of "Assessment of global total column water vapor sounding using a spaceborne differential absorption radar"

_Atmospheric Measurement Techniques, 2020_

## Referee Comment (RC1) · Anonymous Referee #1 · 24 Jun 2020

The paper investigates the value of DAR for retrieving integrated water vapour (IWV). The paper is concise, well written and generally clear. The topic is very important and timely given the recent technology advance for G-band radars. I have few major points that I would like to be addressed.

Major comments:

1) The paper provides a good idea about the performance of the proposed DAR system globally. However the strength/novelty of the methodology to me is to provide IWV in cloudy conditions (in clear sky conditions we can probably be satisfied with current observations), where also I expect to see larger IWV spatial gradients (and so where the

fine resolution of the method could be really useful). So it would be great to see the performances conditioned to cloudy conditions (maybe defined by some LWP thresholds). Also it would be interesting to see a scene (maybe a Stratocumulus or a convective scene from LES) with strong IWV gradients where the retrieval performances can be shown in detail.

2) Just to give an idea to the reader it would be good to know the single-pulse sensitivity for the radar specs tabulated in Tab.1. I expect 30 dB difference between the different powers? Is that correct? Is there any issue with the dynamic range of the surface reflectivity measurements?

3) For Multiple scattering you state: "In all scenarios simulated here, the surface return dwarfed the multiple-scattered component of clouds and rain." Well I am sure this is true everywhere but in deep convection. CloudSat surface return sometime is indeed dwarfed by multiple scattering in deep convection (several examples are provided in literature, e.g. Battaglia and Simmer, IEEE TRANSACTIONS ON GEOSCIENCE AND REMOTE SENSING,VOL. 46, NO. 6, JUNE 2008) I am sure that, when increasing the frequency, such instances will be more. It would be good this is quantified (maybe having a scene like suggested at 1) could help). Also what do the authors mean with "coarse" resolution of Cloudsat hydrometeors ?(I am still confused why the authors need to under-sample Cloudsat (computational time?)

Minor comments: 1) In the abstract I do not think that the auhors actually mean "pulses will reach the surface" (for radar the pulses must also go back to the receiver to be detected!) 2) Sect.3: not clear what scattering model has been used for ice. 3) Sect.3: "we only used every 50 CloudSat measurement" (you mean one out of 50?)

---

## Referee Comment (RC2) · Anonymous Referee #2 · 24 Jun 2020

**General comments:**

This manuscript described a feasibility study of a differential absorption radar operating near a water vapor absorption line (183 GHz) in measuring total column water vapor from space. The considered radar system combined two tones (167 & 174.8 GHz) near the vapor absorption line and had potential radar powers ranging from 0.1 to 100 W. The radar system performances including measurement approaches and environment conditions were numerically simulated based on basic radar principles and global CloudSat observed hydrometeor and ECMWF atmospheric temperature, pressure and humidity profiles. Various measurement sensitivities and uncertainties were simulated.

Results showed that with 20 W radar transmitted power satisfied measurements could be obtained for both land and ocean areas. This kind of satellite measurements could potentially provide additional global water vapor observations, particularly over land, besides passive infrared and microwave vapor soundings over oceans. This study was straight forward, and the manuscript was basically written well.

Based on the importance of water vapor observations, especially over global rural land and polar regions, for weather and climate studies, and the general and specific comments listed below, a minor revision is recommended.

1. The authors should clarify certain simulation procedures in obtaining the simulated random and systematic errors of total column water vapor measurements. Were those errors obtained based on uncertainty parameterization like Eq. 6 or the detailed radar signal propagation processes? Did they simulated the processes of the radar signal generation, transferring through the atmosphere, and reflecting at the surface with all adequate noise and uncertainties added in individual parts of the signal propagation processes? For example, when passing through atmosphere, what turbulence was considered for radar signals?

2. For the radar system considered, what was the swath of spaceborne radar? The NRCS of surfaces may drop with increased scanning angle quickly. Could it scan like the precipitation radar onboard TRMM satellite? A related question is the sampling rate (or revisit time period) for a particular location.

Specific comments:

There are some editing issues throughout the manuscript. Thorough proof-reading is required. 1. Abstract, Line 14:, '.. a fractional yield better than 0.7'. It is not clear what exactly this 'yield' means here. The authors defined this at a very late stage.

2. Pg2, Line 8: add 'a' after 'using'.

3. Pg2, Line 9: add 'for' before 'all surfaces' It would be better if change 'all surfaces'
to 'all surface types' .

4. Pg2, Line 22: The authors wrote: '.... absorption line, respectively...' It would improve the readability if moving the 'respectively' to the end of this statement.

5. Pg2, Line 25: It sounds like there are some issues with the assumption of '0.16 dBZ radar precision, and around a -30dBZ minimum detectability'. Could the authors provide details on these issues, please?

6. Section 2, pg 3 and 4: this part could be shortened because of many cited papers and previous studies.

7. Pg3, L14: add 'parameter' after 'where C(ïĄő) is the radar system'.

8. Pg4, L1: No need to make subsection 2.1. All in Section 2 would be fine.

9. Pg4, L12 and 25: The authors used a 66-us chirped pulse. This is a pretty long pulse. How big are the sidelobes of the pulse returns after coherent integration (or correlator)? Could the radar backscatterers at different ranges affect each other? What is the impact of those sidelobes? For example, what is the potential bias these sidelobes could produce when rain drops are considered?

10. Pg4, L20: The authors mentioned NT value is, at least, 2. Is it possible for the designed radar system to transmit the two tones together? If yes, what is the potential of cross-talks? If not, provide reasons besides cross-talks due to transmission and amplification.

11. Pg4, L26: change 'table' to 'Table' for consistency. Also, please check other places such as line 7 in page 5.

12. Fig. 1: Need enlarge the symbols of half circle and triangle. They are hard to read now.

13. Pg5, L14: What was the cited reference 'Partin 2007' exactly? A thesis or internal report? In either case, a link to the document is needed.

AMTD
14. Pg5, L16: For radar operation frequencies of 167 and 174.8 GHz, could the authors let readers know how big the atmospheric gaseous attenuations at these frequencies are for some typical clear atmospheric profiles such as those with total column water vapor values of 35, 45 and 55 mm? What are the percentages of vapor attenuations on the totals?

15. Pg5, L18 to 21: The study used two vapor profiles: wet and dry. It seems that the dry profile was not for zero vapor amount. Could the authors let readers know what the total column water vapor values were used in the simulation for wet and dry conditions? Also, how about other meteorological conditions such as the surface temperatures?

16. Fig. 3: the colors in the figure were not easy to read due to inconsistency from cold to warm colors. The authors need change the color code.

17. Pg6, L5 to 8: only 4 panels were shown. Please check this statement. Also, this statement may be too complicated and should be split into shorter sentences.

18. Eq. 17: Please provide references or a brief derivation to obtain this equation. Was the uncertainty used in the manuscript a variance or mean square error? What assumptions did the authors used in deriving this equation? This was confusing since it was not clear if the means or bias errors were included.

19. Pg7, L8: no need of the subsection 4.1.

20. Pg7, L15: '.... also provided by CloudSat.' Did the authors mean '... also provided by CloudSat-ECMWF product.'

21. Pg7, L17: Wi was a water vapor state variable. Readers expected it to be a vector for the water vapor profile. However, the iteration parameter ïĄij1 ïĂ■ Wi+1 / Wi ïĄij made it looked like a scaler. Was this value the total column water vapor? Please clarify this.

22. Pg7, L19 to 24 and Eqs. 19 and 20: Were instrument and measurement noises added when calculating the simulated radar returns?

AMTD
23. Pg8, L2: The authors mentioned precision here. Could the authors clarify if the retrievals had bias errors when only instrument and measurement noises were used? Many factors could introduce biases. For example, as mentioned previously, sidelobes could cause bias errors. If the answer is yes, how small or big are these biases?

24. Pg8, L4: '.. the impact of not knowing of temperature and pressure by using climatological values' How could this happen? People would think the authors or users of the spaceborne radar measurements should have products of numerical weather forecasts, assimilations and/or analyses of these temperature and pressure profiles? It is understandable to have certain uncertainties (or random and bias errors) associated with these modelled values, but it seems to pretty extreme to think without information on these values during environmental science satellite operations.

25. Pg8, L15 to 17: Could the authors move the discussion on Fig 5 (SNR) after Fig. 4 discussions. That is, move these lines to the end of line 21.

26. Pg8, L18: The authors defined 'yield' here. For increasing readability, it should be defined much earlier when the first time it was used.

27. Pg8, L19 and 20: change the words 'used before' to 'as those shown in left panels'

28. Pg9, L9: define 'pT'.

29. Pg9, L25 and 26: The authors cited Meshkov (2006). The reference showed that this was a thesis. There was an article with the same title by Meshkov and De Lucia (2005). Were the essential contents of these two articles the same? If yes, the authors should cite the latter because of easier to obtain for readers.

30. End of pg9 and beginning of pg10: The authors found that potentially current uncertainties (4%) in the line width of the water vapor absorption line could cause about 1.4 mm total column water vapor bias errors. For this kind of significant systematic errors, can calibration and validation of the measurements of the instrument or even AMTD
using an airborne radar at the frequencies considered over tropical regions or during midlatitude humid summer periods identify the bias and correct this potential systematic error? From random error analysis, it seems possible with long enough averages. If yes, the authors should make some comments and explanations, especially at the summery, on this, which would increase the feasibility of the instrument.

---

## Author Comment (AC1) · 2 Jul 2020

We thank the reviewer for her/his comments. We note that the reviewer comments refer to a previous version of the document. Those comments were already addressed. Below are our *previous* responses to them in blue.

General comments:

This manuscript described a feasibility study of a differential absorption radar operating near a water vapor absorption line (183 GHz) in measuring total column water vapor from space. The considered radar system combined two tones (167 & 174.8 GHz) near the vapor absorption line and had potential radar powers ranging from 0.1 to 100 W. The radar system performances including measurement approaches and environment conditions were numerically simulated based on basic radar principles and global CloudSat observed hydrometeor and ECMWF atmospheric temperature, pressure and humidity profiles. Various measurement sensitivities and uncertainties were simulated. Results showed that with 20 W radar transmitted power satisfied measurements could be obtained for both land and ocean areas. This kind of satellite measurements could potentially provide additional global water vapor observations, particularly over land, besides passive infrared and microwave vapor soundings over oceans. This study was straight forward, and the manuscript was basically written well.

Based on the importance of water vapor observations, especially over global rural land and polar regions, for weather and climate studies, and the general and specific comments listed below, a minor revision is recommended.

1. The authors should clarify certain simulation procedures in obtaining the simulated random and systematic errors of total column water vapor measurements. Were those errors obtained based on uncertainty parameterization like Eq. 6 or the detailed radar signal propagation processes? Did they simulated the processes of the radar signal generation, transferring through the atmosphere, and reflecting at the surface with all adequate noise and uncertainties added in individual parts of the signal propagation processes? For example, when passing through atmosphere, what turbulence was considered for radar signals?

We believe that this is well explained through the manuscript, in particular section 3 (radar instrument simulator). The first paragraph states: Radar returns are simulated using the radiometric model described in Millán et al. (2014) and Millán et al. (2016). In short, radar reflectivities are estimated using the time-dependent two-stream approximation (Hogan and Battaglia, 2008), gaseous absorption is evaluated using the clear sky forward model for the EOS Microwave Limb Sounder (Read et al., 2004), hydrometeors scattering properties are evaluated using Mie scattering theory (assuming spherical hydrometeors), and the surface cross section is calculated using a quasi-specular scattering model (Li et al., 2005) for the ocean surface.

Turbulent process occurring at scales smaller than the CloudSat foorprint were not included. We added: It is noted that non-uniformities within the beam at scales smaller than the CloudSat horizontal resolution are not included in these simulations. They will be better addressed using either high resolution ground-based data or high resolution atmospheric models in subsequent studies.

To clarify how we studied the impact of the measurement noise we added (in section 4) the following: Instrument noise is not added to any of these simulations because its impacts are studied through equation 16.

2. For the radar system considered, what was the swath of spaceborne radar? The NRCS of surfaces may drop with increased scanning angle quickly. Could it scan like the precipitation radar onboard TRMM satellite? A related question is the sampling rate (or revisit time period) for a particular location.
The instrument simulated does not perform any cross-track scanning. This is an important issue to consider, however one could interpret these nadir calculations in an off-nadir context by scaling the angle dependence of the radar cross section with the various transmit powers that we consider here.

Specific comments:

There are some editing issues throughout the manuscript. Thorough proof-reading is required.

1. Abstract, Line 14:, '.. a fractional yield better than 0.7'. It is not clear what exactly this 'yield' means here. The authors defined this at a very late stage.
The text was changed to: Results indicate that, using two radar tones at 167 and 174.8GHz with a transmit power of 20W ensures that both pulses will reach the surface at least 70% of the time in the tropics and more than 90% of the time outside the tropics, and that total column water vapor can be retrieved with a precision better than 1.3 mm.

2. Pg2, Line 8: add 'a' after 'using'  Done

3. Pg2, Line 9: add 'for' before 'all surfaces' It would be better if change 'all surfaces' to 'all surface types'  Done

4. Pg2, Line 22: The authors wrote: '.... absorption line, respectively…' It would improve the readability if moving the 'respectively' to the end of this statement.  Done

5. Pg2, Line 25: It sounds like there are some issues with the assumption of '0.16 dBZ radar precision, and around a -30dBZ minimum detectability'. Could the authors provide details on these issues, please?
The text was changed to: However, both studies assumed a *constant* instrument error model, with a 0.16dBZ radar precision, and around a -30dBZ minimum detectable signal.  Here we use a more realistic uncertainty model (described in section 2) that includes speckle noise. That is, that depends on the magnitude of the return power which in turn depends on the water vapor burden.

6. Section 2, pg 3 and 4: this part could be shortened because of many cited papers and previous studies.
We decided to leave it as is for completeness, that way the reader can understand the paper without having to read the cited papers.

7. Pg3, L14: add 'parameter' after 'where C is the radar system'.  Done

8. Pg4, L1: No need to make subsection 2.1. All in Section 2 would be fine.  Done

9. Pg4, L12 and 25: The authors used a 66-us chirped pulse. This is a pretty long pulse. How big are the sidelobes of the pulse returns after coherent integration (or correlator)? Could the radar backscatterers at different ranges affect each other? What is the impact of those sidelobes? For example, what is the potential bias these sidelobes could produce when rain drops are considered?  We added the following sentence at the end of that paragraph: Despite the relatively long chirp time, we do not simulated any sidelobes because, as demonstrated by RainCube (a Ka-band 6U cubesat radar with a 166-us pulse length), through an optimal selection of pulse shape and digital processing, sidelobes can be suppressed to accurately measure the most relevant precipitation processes near the surface [Peral et al (2019)].

10. Pg4, L20: The authors mentioned NT value is, at least, 2. Is it possible for the designed radar system to transmit the two tones together? If yes, what is the potential of cross-talks? If not, provide reasons besides cross-talks due to transmission and amplification.
Although theoretically it may be possible to transmit the two tones together, as simulated (and probably as implemented in a real instrument due to practicality) the radar system will not likely transmit the two tones together. Other cross-talk issues, if any, will have to be studied and corrected in the actual radar implementation, but in these simulations, they were not studied.

11. Pg4, L26: change 'table' to 'Table' for consistency. Also, please check other places such as line 7 in page 5.
Done

12. Fig. 1: Need enlarge the symbols of half circle and triangle. They are hard to read now.   Done

13. Pg5, L14: What was the cited reference 'Partin 2007' exactly? A thesis or internal report? In either case, a link to the document is needed. Internal document, the reference was updated.

14. Pg5, L16: For radar operation frequencies of 167 and 174.8 GHz, could the authors let readers know how big the atmospheric gaseous attenuations at these frequencies are for some typical clear atmospheric profiles such as those with total column water vapor values of 35, 45 and 55 mm? What are the percentages of vapor attenuations on the totals?

We included a new figure:

[Figure]

At the end of that paragraph we added: As examples of this burden, Figure 2 shows the spectral variation of the surface return for different total column water vapor burdens under clear sky conditions. The spectral contrast between 174.8 and 167GHz varies from 0.1dB for no water vapor to 31dB for 60mm of total column water vapor.

15. Pg5, L18 to 21: The study used two vapor profiles: wet and dry. It seems that the dry profile was not for zero vapor amount. Could the authors let readers know what the total column water vapor values were used in the simulation for wet and dry conditions? Also, how about other meteorological conditions such as the surface temperatures?
We added the following: (with 44 and 4mm of total column water vapor burden and 298 and 257K surface temperature, respectively)

16. Fig. 3: the colors in the figure were not easy to read due to inconsistency from cold to warm colors. The authors need change the color code.
We changed the colorbar for panels b and c (the panels showing the simulated CloudSat-driven effective surface cross-section for both frequencies). The colorbars for a and d are the same (but reverse or inconsistent as the reviewer points out) on *purpose* to highlight the similarities between the total column water vapor (panel a) and the difference between the two surface cross sections (panel d).

17. Pg6, L5 to 8: only 4 panels were shown. Please check this statement. Also, this statement may be too complicated and should be split into shorter sentences.

*The statement was broken into 3 sentences (and the number of panels was corrected): The top panel is an 8-day average (January 1-8, 2007) of the CloudSat ECMWF-aux total column water vapor to show the context of the simulations. The next two panels show the average effective surface cross section, that is sigma0 at 167 and 174.8GHz. Lastly, the bottom panel shows the difference between the 174.8 and 167GHz simulations.*

18. Eq. 17: Please provide references or a brief derivation to obtain this equation. Was the uncertainty used in the manuscript a variance or mean square error? What assumptions did the authors used in deriving this equation? This was confusing since it was not clear if the means or bias errors were included.

*In the manuscript we added after that equation:  which is simply the propagation of the individual errors of Pr(v2) and Pr(v1).*

19. Pg7, L8: no need of the subsection 4.1.   *Done*

20. Pg7, L15: '…. also provided by CloudSat.' Did the authors mean '… also provided by CloudSat-ECMWF product.'

*Yes, It was changed to that.*

21. Pg7, L17: *Wi* was a water vapor state variable. Readers expected it to be a vector for the water vapor profile. However, the iteration parameter 🮐1 − _Wi+1 / Wi 🮐_ made it looked like a scaler. Was this value the total column water vapor? Please clarify this.

*As specified in line 1 of page 7 (of the original manuscript):  the shape of the assumed water vapor profile does not change during the retrieval, it is simply scaled according to the total column water vapor retrieved.*

22. Pg7, L19 to 24 and Eqs. 19 and 20: Were instrument and measurement noises added when calculating the simulated radar returns?   *No, measurement noise is studied through equation 16 as it is often used when doing this type of studies. We could equivalently add the noise to the observed radar powers however it will not change any conclusions. The following sentence was added: Instrument noise is not added to any of these simulations because its impacts are studied through equation 16.*

23. Pg8, L2: The authors mentioned precision here. Could the authors clarify if the retrievals had bias errors when only instrument and measurement noises were used? Many factors could introduce biases. For example, as mentioned previously, sidelobes could cause bias errors. If the answer is yes, how small or big are these biases?

*No bias errors were investigated in this paper. Those biases are diagnosed using the mean value returned by an* actual *retrieval versus the* truth state *(given by in-situ measurements for instance). Those biases will vary depending on the actual retrieval scheme implemented, number of iterations, linearization profile, temperature and pressure used, etc, and hence not included in this study. The purpose of this study is to evaluate the possible precision and systematic biases and those ones, should not vary depending on the retrieval specifics.*

24. Pg8, L4: '.. the impact of not knowing of temperature and pressure by using climatological values' How could this happen? People would think the authors or users of the spaceborne radar measurements should have products of numerical weather forecasts, assimilations and/or analyses of these temperature and pressure profiles? It is understandable to have certain uncertainties (or random and bias errors) associated with these modelled values, but it seems to pretty extreme to think without information on these values during environmental science satellite operations.

*The reviewer is correct, in any retrieval scheme products of numerical weather forecast will be used as part of the retrieval. However, we used climatological values versus reanalysis values to study the* worst possible *impact of the uncertainties (as the reviewer also points out) of the modelled values. We changed that phrase to:  … the impact of not knowing* precisely *the temperature and pressure …*

Further, in the bullet describing these errors we added: Simply, these errors evaluate the worst possible impact of not knowing precisely the temperature and pressure.

25. Pg8, L15 to 17: Could the authors move the discussion on Fig 5 (SNR) after Fig. 4 discussions. That is, move these lines to the end of line 21.
After consideration, we believe that the text as is reads better, since the discussion of Figure 5 ties directly with: Further increasing the transmit power does little to improve the precision, because as we noted in section 2.1, in high SNR regimes, the fractional error in the measurement is largely determined by the number of uncorrelated pulses used.

26. Pg8, L18: The authors defined 'yield' here. For increasing readability, it should be defined much earlier when the first time it was used. The text in the abstract was changed, now, yield is defined immediately after its first mention.

27. Pg8, L19 and 20: change the words 'used before' to 'as those shown in left panels'   Done

28. Pg9, L9: define 'pT'. We added (pressure-temperature)

29. Pg9, L25 and 26: The authors cited Meshkov (2006). The reference showed that this was a thesis. There was an article with the same title by Meshkov and De Lucia (2005). Were the essential contents of these two articles the same? If yes, the authors should cite the latter because of easier to obtain for readers.
We changed the citation to the article as recommended

30. End of pg9 and beginning of pg10: The authors found that potentially current uncertainties (4%) in the line width of the water vapor absorption line could cause about 1.4 mm total column water vapor bias errors. For this kind of significant systematic errors, can calibration and validation of the measurements of the instrument or even using an airborne radar at the frequencies considered over tropical regions or during midlatitude humid summer periods identify the bias and correct this potential systematic error? From random error analysis, it seems possible with long enough averages. If yes, the authors should make some comments and explanations, especially at the summery, on this, which would increase the feasibility of the instrument.
We added the following text: However, this type of bias should be easily corrected during a validation campaign since all retrievals will be off by the same *constant* amount.

---

## Author Comment (AC2)

We thank the reviewers for her/his comments. Below are our responses in blue.

The paper investigates the value of DAR for retrieving integrated water vapour (IWV). The paper is concise, well written and generally clear. The topic is very important and timely given the recent technology advance for G-band radars. I have few major points that I would like to be addressed.

Major comments:

1) The paper provides a good idea about the performance of the proposed DAR system globally. However the strength/novelty of the methodology to me is to provide IWV in cloudy conditions (in clear sky conditions we can probably be satisfied with current observations), where also I expect to see larger IWV spatial gradients (and so where the fine resolution of the method could be really useful). So it would be great to see the performances conditioned to cloudy conditions (maybe defined by some LWP thresholds). Also it would be interesting to see a scene (maybe a Stratocumulus or a convective scene from LES) with strong IWV gradients where the retrieval performances can be shown in detail.

Our simulations include many cloudy and precipitating scenes. To make this clearer in page 6 line 16 after, "In these maps there are around 80,000 simulations (we only used every 50 CloudSat measurements)." we will add: These simulations include, according to the CloudSat classification algorithm [Sassen and Wang, 2008] more than 10,000 clouds identified as cirrus and stratocumulus, around 500 identified as Cumulus, 7000 as nimbostratus, and 800 as deep convection. Further, these simulations include around 400 precipitating scenes with rain rates of up to 4.5 mm hr-1 according to the rain profile product [L'Ecuyer and Stephens, 2002].

The impacts of clouds can clearly be scene in Figure 7 which shows how the yield is affected under clear sky conditions versus all sky conditions. In page 9 line 4, We will add the following: The yield, however, improves substantially. For example, in the tropics, for 20W of transmit power, the yield becomes better than 0.85 (as opposed to better than 0.7) and for 50W become better than 0.95 (as opposed to better than 0.8). *This yield improvement under clear sky cases is due to the lack of the attenuation burden impose by hydrometeors.*

At this point we will add the following figure:

With the accompanying text: To further highlight that this technique will work under cloudy and precipitating conditions, Figure 8 shows a cross section of CloudSat-driven simulations over the Southern Ocean. This cross section consists of 500 CloudSat profiles encompassing ice clouds, liquid clouds, rain, and snow. Yields in this cross section are similar to those shown in Figure 7 at around 55S for the all scenes zonal average yield.

This new figure will have the following caption: Cross section exemplifying the CloudSat-driven simulations (data from 1 January 2007 over the Southern Ocean). (a) Simulated CloudSat-driven radar reflectivity at 167 GHz. (b) CloudSat retrieved total (IWC+LWC+rain+snow) hydrometeor water content. Black and red lines delimit areas where snow and rain were detected. (c) ECMWF-aux water vapor. (d) Total column water vapor (black solid line) as well as the retrieval precision (dashed lines) for different transmit powers and locations where at least one of the radar pulses was attenuated beneath the noise floor. Yield values for each simulated transmit powers are given by the numbers in brackets.

We did not use LWP because it breaks down under rainy condition where the CWC-RO algorithm fails. But we believe panel (b) shows clearly the hydrometeor burden.

Further, to emphasize that the *clear sky* systematic uncertainty (section 5.2) is really talking about cloudy and precipitating scenes we will change its name to "Clouds and precipitation errors" and the figure legend to "Cloud and Precip".

To emphasis that the method will work in cloud and precipitation regions, the explanation of the systematic errors will be expanded to (page 10 line 7): As shown in Figure 8, most of the potential systematic uncertainties are lower than 0.5 mm, *including those uncertainties accounting for the extra attenuation impose by clouds and precipitation (as long as they do not attenuate completely the radar pulses)*. The exception are the errors associated with H2O 183 GHz line width which could be as big as 1.4 mm...

Sassen, K., and Z. Wang, (2008) Classifying clouds around the globe with the CloudSat radar: 1-year of results, Geophys. Res. Lett., 35, L04805, doi:10.1029/2007GL032591

L'Ecuyer, T. S., and G. L. Stephens, 2002: An estimation-based precipitation retrieval algorithm for attenuating radars., J. Appl. Meteor., 41, 272-285.

2) Just to give an idea to the reader it would be good to know the single-pulse sensitivity for the radar specs tabulated in Tab.1. I expect 30 dB difference between the different powers? Is that correct? Is there any issue with the dynamic range of the surface reflectivity measurements?

We will add to table 1 the minimum detectable  $dB(s_0T^2)$  for each of the transmit power, the new line will say:

Minimum detectableb dB( $s_0T^2$ ) -18, -28, -38, -41, -45, -48. bfor each of the transmit powers considered, respectively.

We do not really know what the reviewer means by dynamic range of the surface reflectivity, it is precisely that range, the difference between 167 and 174.8 return the signal that we are exploiting to retrieve total column water vapor.

3) For Multiple scattering you state: "In all scenarios simulated here, the surface return dwarfed the multiplescattered component of clouds and rain." Well I am sure this is true everywhere but in deep convection. CloudSat surface return sometime is indeed dwarfed by multiple scattering in deep convection (several examples are provided in literature, e.g. Battaglia and Simmer, IEEE TRANSACTIONS ON GEOSCIENCE AND REMOTE SENSING,VOL. 46, NO. 6, JUNE 2008) I am sure that, when increasing the frequency, such instances will be more. It would be good this is quantified (maybe having a scene like suggested at 1) could help). Also what do the authors mean with "coarse" resolution of Cloudsat hydrometeors ? (I am still confused why the authors need to undersample Cloudsat (computational time?)

The multiple scattering discussion on page 10 line 11 will be change to: In all scenarios *simulated here*, the surface return dwarfed the multiple-scattered component of clouds and rain. That is, the systematic uncertainty induced by ignoring multiple scattering effects was negligible, because the screening of the precipitating scenes (disregarding profiles which had any negative values) screened out those scenarios where multiple scattering was present. However, we do not anticipate that multiple scattering will be a problem, because according to Battaglia et al 2008, 80% of the rainy profiles can be accurately modeled assuming a single scattering approximation, and, further, in the order 20% of the cases, the strong hydrometeor burden will hinder the surface return.

We do under-sample CloudSat to save computational time, this will be stated clearer. In page 5 line 14, after the sentence "Note that, to decrease the number of calculations we subsample these fields, we only used one out of 50 CloudSat measurement.", we will add: In other words, we under-sample CloudSat to save computational time.

Battaglia et al (2008), Identifying multiple-scattering-affected profiles in CloudSat observations over the oceans, doi: 10.1029/2008JD009960

**Minor comments:**

1) In the abstract I do not think that the authors actually mean "pulses will reach the surface" (for radar the pulses must also go back to the receiver to be detected!)

The reviewer is absolutely correct, we will change to: ...both pulses will be detected with a signal to noise ratio > 1 at least 70% ...

In page 8 line 25 where we define yield, the sentence will also be changed to: number of times surface reflections at both frequencies are detected with an SNR > 1 divided by the total number of simulations.

**2) Sect.3: not clear what scattering model has been used for ice.**

As mention in line 5 of page 5 (first paragraph of that section), "we use Mie scattering theory assuming spherical solid hydrometeors", that is, we evaluate ice, liquid cloud particles, rain and snow using Mie theory. If the reviewer is asking for the particle size distribution that is given in table 2. Currently (at the end of that paragraph) it just states, More details can be found in Table 2. To make it clearer, this last sentence will be changed to: More details, *such as the dielectric constants and the particle size distributions used*, can be found in Table 2.

**3) Sect.3: "we only used every 50 CloudSat measurement" (you mean one out of 50?)**

Yes, we mean one out of 50, the sentence will be changed to: Note that, to decrease the number of calculations (save computational time) we subsample these fields, we only used one out of 50 CloudSat measurement.